# The Mental Health Impact of the COVID-19 Pandemic on Older Adults in China: A Systematic Review

**DOI:** 10.3390/ijerph192114362

**Published:** 2022-11-02

**Authors:** Jingyuan Liu, Crystal Kwan, Jie Deng, Yuxi Hu

**Affiliations:** Department of Applied Social Science, Hong Kong Polytechnic University, Hong Kong, China

**Keywords:** mental health, COVID-19 pandemic, older adults, China, systematic review

## Abstract

Considered at a high risk during the COVID-19 pandemic, older adults in China not only face the disadvantages caused by their relatively low immune systems, but also the challenges brought about by the complex psychological environment in which they spend this special period of their life. However, a thorough study on the impact of the pandemic on older adults’ mental health in China remains scant. Hence, this research aimed to investigate the question: What are the mental health outcomes and associated risk factors of the COVID-19 pandemic on older adults in China? Two Chinese academic databases (China National Knowledge Infrastructure and WANFANG DATA) as well as six English academic databases (PubMed, Scopus, Web of Science, MEDLINE, Social Science, and Google Scholar) were searched while following PRISMA guidelines. Studies were selected according to the predetermined inclusion criteria. Further, relatively high detective rates of mental health disorders, including anxiety symptoms (4.9% to 48.6%), depression symptoms (13.8% to 58.7%), hypochondria (11.9%), suicidal ideation (4.1%), along with worries and fear (55.7%) were all reported. The COVID-19 pandemic has presented a threat to not only the physical, but also the psychological health of Chinese older adults. The most common risk factors of psychological distress among Chinese older adults were found in female gender, living in rural areas, coexisting chronic diseases, and insufficient knowledge about the COVID pandemic. As a result, government policy and psychological guidelines that are created in order to alleviate the adverse effects of COVID-19 on older adults’ mental health, need to be further developed.

## 1. Introduction

The novel coronavirus (COVID-19) was first identified by the World Health Organization (WHO) in January 2020. Later in March 2020, the spread of COVID-19 was declared a global pandemic [1]. Since the beginning of the 21st century, many epidemics caused by viruses (such as SARS and MERS) have circulated. However, COVID-19 has been observed to have spread at the highest speed and caused an enormous number of deaths [2]. To prevent or stop the spread of the COVID-19 infection, measures—including stopping production in non-essential business sectors—quarantines, and lockdowns have all been undertaken by authorities in almost all countries around the world [3].

The worries and uncertainties associated with the outbreak of the pandemic, along with the states of isolation that are caused by mass lockdowns and quarantines can be psychologically distressing for anyone during the pandemic [4]. Older adults, however, may be especially vulnerable to the adverse mental health effect of the pandemic. On one hand, due to their effete immune systems and concomitant chronic diseases, older adults are considered most at risk of severe illness from COVID-19 [2]. Secondary bacterial infection and respiratory failures are frequently reported in older adults [5]. Additionally, while the mortality rate is around 2% to 3% for healthy adults, the propensity is three times higher for older adults [6]. In Shanghai China, during the pandemic, more than 90% of COVID-related deaths were found in older adults [7]. On the other hand, due to the mass lockdowns and quarantines, older adults have also been deprived of the help and social support they would normally receive from their caregivers and families [8]. Many older adults may not be skillful at technologies (e.g., smart phones and laptops), which may lead to greater emotional distancing as there are without even digital contact from their loved ones. Moreover, the social stigma of ageism is also starkly revealed by this outbreak, which has led to increased elder abuse and marginalization [6,9]. All these factors may accumulate stress and fear among older adults, and, thus, bring more challenges to their psychological conditions under the impact of the pandemic. Therefore, China, a country with the largest aging population globally [10], should pay more attention to older adults’ mental health needs in the context of the normalized prevention and control of the COVID-19 pandemic.

Existing systematic reviews and meta-analyses have investigated the effect of the COVID-19 pandemic on the mental health of the Chinese general population [11] and children [12]. The only systematic review that has examined the mental health status of Chinese older adults during the pandemic has focused on the prevalence of depression and anxiety exclusively [13]. To our knowledge, this is the first review that investigates the influence of the pandemic on Chinese older adults’ mental health comprehensively. The objective of this review, therefore, is to summarize the published literature that has reported on older adults’ mental health outcomes during the COVID-19 pandemic and to then identify the associated risk and protective factors.

## 2. Method

This review (protocol registered on PROSPERO-CRD42022310212) was conducted based on the preferred reporting items for systematic review and meta-analyses (PRISMA) guidelines [14].

### 2.1. Search Strategy and Selection Criteria

A comprehensive search strategy was conducted by three independent researchers (LJ, DJ, and HXY) in order to identify all relevant published studies. This search included two sources, which were the electronic databases and reference lists that contained the included studies. Two Chinese academic databases (China National Knowledge Infrastructure and WANFANG DTA) and six English academic databases (PubMed, Scopus, Web of Science, MEDLINE, Embase, and Google Scholar) were searched in order to identify relevant studies. Identified studies were screened based on the title and abstract, and, after excluding obviously irrelevant articles, the full texts of the remaining studies were examined in order to determine compliance with the inclusion criteria.

The search terms were selected based on common synonyms used for each of the main terms, which were: COVID, Chinese, older adults, and mental health. Related words, synonyms, and plurals of the main terms were selected according to the search results in databases, such as PubMed. The team also identified synonyms or related words based on other published systematic reviews that focused on the mental health impact of COVID-19 [15]. The COVID terms included COVID* OR COVID-19 OR Coronavirus. The mental health terms included mental health* OR mental disorders OR psychological stress OR psychiatric symptoms OR mental illness. The older adults’ terms included elderly* OR senior OR older adults* OR older people OR older persons OR aging population OR aged people.

### 2.2. Condition and Individuals Being Studied

The inclusion criteria were as follows: (1) There are no restrictions on the type of study design (i.e., qualitative, quantitative, and mixed method designs were all eligible); (2) the studies investigated included mental health disorders and associated risk factors relevant to the COVID-19 pandemic; (3) aligning with the definition of the United Nations (2017), populations were restricted to older people (age ≥ 60 years old) in the region of China only; (4) the studies reviewed were either in Chinese or English only; and (5) the studies considered were from the January 2020 (the start of the pandemic) to April 2022 period. The exclusion criteria were as follows, studies: (1) Whereby data cannot be extracted based on age group; (2) in any language other than Chinese and English; (3) with unavailable full text; and (4) were not peer reviewed. The search was conducted from March to April 2022.

### 2.3. Data Extraction

Independent investigators (LJ and DJ) carried out data extraction. Any discrepancies that arose were resolved through consensus and through consulting another investigator (CK) if an agreement was not attained. Selected variables that were extracted included: journal information (authors; year of publication), study design (e.g., cross-sectional, qualitative, mixed methods, etc.), sample size, participants’ characteristics (e.g., sex, age, marital status, education, income, residing in urban vs. rural, and COVID infectious status), and key findings related to mental health.

### 2.4. Risk of Bias (Quality) Assessment

Quality assessment was conducted by using the appropriate respective appraisal tool for each research design. Two investigators (LJ and CK) independently assessed the methodological quality of cross-sectional studies using the JBI cross-sectional analytical checklist [16]. The other two investigators (DJ and HYX) independently assessed the methodological quality of mixed-method studies using the Mixed Methods Appraisal Tool [17]. Additionally, the quality of the cohort study included was assessed by two investigators (DJ and HYX), independently, using the JBI checklist for cohort study. Results of these checklists are attached as Appendix A.

### 2.5. Strategy for Data Synthesis

A parallel-results convergent design was planned, which consisted of independent syntheses of qualitative and quantitative evidence, as well as an interpretation of the results in the discussion. Since the studies included were heterogenous, synthesis without meta-analysis (SWiM) by [18] was planned for the quantitative findings. There were no qualitative studies and only one mixed-method study. The quantitative findings of the mixed-method study were extracted and synthesized with findings of other quantitative studies. The integration of qualitative and quantitative findings was guided by [19] whereby the aims were to: (i) identify common concepts across both sets of findings; (ii) “develop” side-by-side comparisons of the quantitative and qualitative results pertaining to the concepts; and (iii) determine, through a narrative discussion, in what ways they confirm, disconfirm, or expand on each other.

## 3. Results

### 3.1. Search Results

In total, 586 studies were identified. Of those, 384 studies remained after excluding duplicates. A further 336 studies were removed after the initial screening of titles and abstracts. As such, a total of 48 full-text studies were assessed for eligibility. A further 4 studies were excluded due to the fact that the sample included other age groups and the data related to the older age group (age ≥ 60) could not be separated in the findings. Another 15 studies were excluded as the populations that were studied were not specific to older adults in China. Lastly, 9 studies were excluded because the main outcomes were not mental health related. Following the full-text screening, 20 studies met the inclusion criteria. Figure 1 illustrates the identification of eligible studies.

### 3.2. Study Characteristic

Study characteristics and study findings related to mental health are summarized in Table 1. The sample size of the 20 studies ranged from 49 [20] to 6467 [21] participants, with a total of 14,578 participants. Most studies followed a cross-sectional design (*n* = 18), while a cohort study and mixed-method study are also included. In 19 studies, participants completed self-reported surveys for data collection; moreover, in only two studies, were face-to-face interviews utilized (including the mixed-method study that used both self-reported surveys as well as face-to-face interviews). Most studies (*n* = 18) investigated older adults from a general population—one study focused on empty nesters [22], whereas another one on COVID-19 infected older adults [23]. A variety of scales (*n* = 16) were used in the studies for assessing different mental health outcomes.

The mental health related outcomes varied across the studies. A total of 9 studies included measures of depression symptoms, while 10 studies included measures of anxiety. Symptoms of hypochondria, suicidal ideation, and other psychological impacts (e.g., fear, worry, neurasthenia, sense of alienation, etc.) of COVID-19 were evaluated in 6 studies. It was additionally observed that one study did not explicitly report mental distress; this is notwithstanding that associated protective factors were identified and discussed.

### 3.3. Quality Appraisal

The results of the study quality appraisal are attached as a Appendix A. The overall quality of the studies is moderate. From 18 of the cross-sectional studies, 11 studies were regarded as having high quality (positive answers were given to all 7 questions), 6 studies were regarded as having moderate quality (1 negative answer was given out of 7 questions), and 1 study was regarded as having low quality (2 or more negative answers were given out of 7 questions). Additionally, the quality of the cohort study was considered high, and the mixed-method study considered moderate.

### 3.4. Symptoms of Anxiety and Associated Factors

The most commonly evaluated mental health outcome was anxiety (*n* = 10), with the prevalence varying from 4.9 % to 48.6% between studies [21,22,23,24,29,31,32,34,36,37].

Symptoms of anxiety exacerbation were reported in two studies [34,36]. One cross-sectional study compared data collected in October 2019–January 2020 with the data collected from the same sample in April–May 2020; further, it was reported that the detection rate of anxiety symptoms in the older adults was 4.95% before the outbreak of COVID-19, and 10.10% during the pandemic (*p* < 0.05) [36]. Similarly, according to Ye and Lin [34], the participants’ (*n* = 340) detection rate of anxiety symptoms had significantly increased from 22.6% to 27.2% (*p* < 0.05), when compared with previous research results before the pandemic. Two studies compared older adults with younger groups. Tao and colleagues [31] compared older adults with university students and found that 32% of older adults reported symptoms of anxiety in the survey using the Self-rating Anxiety Scale (SAS), which was significantly higher than university students (17.5%, *p* < 0.001). In contrast, Su and colleagues [29] found that when compared with middle aged participants (aged 45–59), older adults showed less severe anxiety symptoms.

Many predictive factors of anxiety symptoms were identified in the studies. Firstly, when compared with male older adults, female older adults were reported as more likely to develop symptoms of anxiety [22,34]. Further, insufficient knowledge of the pandemic was also a predictive factor of anxiety symptoms among older adults [24,31,36]. Moreover, as expected, chronic diseases were associated with anxiety symptoms in three studies [24,36,37].

Other associated factors for anxiety symptoms included a lower BMI index, living in rural areas, were quarantined (or people around were quarantined for medical observation), lower monthly income, reporting difficulties in daily life (e.g., in short of daily necessities), and in the obtaining of COVID-19 related information from friends or family members instead of from TV/newspapers [24,36,37]. In particular, a study that focused on empty nesters reported that the SAS scores of empty nesters were significantly higher than the general population (*p* < 0.05), which indicates that living alone should be included when risk factors of anxiety are considered [22].

### 3.5. Symptoms of Depression and Associated Risk Factors

The second most commonly reported negative outcome was symptoms of depression (*n* = 9). The prevalence of depression symptoms varied from 13.8% to 58.7% across studies [22,23,24,26,28,33,34,35,36,38]. Although the reported depression rates during the pandemic are higher than previously investigated, specifically in its prevalence among Chinese older adults [33,34], it is worth noting that the presence of symptoms of depression does not stand for a clinical diagnosis of depression.

The predictive factors of depression symptoms included coexisting chronic disease, lower education level, lack of visits or communication from families, the family members or acquaintances representing a risk to COVID-19, daily viewing of pandemic news for more than 4 h, lower monthly family income, living in rural areas, living alone, and being single or divorced [24,26,33,35,38]. In particular, the cohort study reported that greater depression symptoms at T0 (before the outbreak of the pandemic) and T1 (the beginning of the outbreak) were significantly associated with fewer physical activities at T1 and T2 (after the subsidence of COVID-19). However, the opposite direction of fewer physical activities being associated with symptoms of depression was not identified [28].

### 3.6. Symptoms of Hypochondria, Suicidal Ideation, Insomnia and Other Adverse Mental Health Outcomes

Symptoms of hypochondria were reported with detected rates of 11.9% [24]. Chronic diseases, a lower BMI index, and living in rural areas were found to be associated with greater hypochondria symptoms [24]. Similarly, the qualitative interview results in the mixed method study [38] also stated that older adults with a chronic disease were worried, and also often suspected, that they were already infected with COVID-19. A suicidal ideation rate of 4.1% was reported in Liang and Deng’s [26] study. However, among those participants who experienced suicidal ideation, 31.9% believe there is a need for mental health services, yet only 10.6% had reached out for help. The insomnia symptoms were examined in one study that focused on older adults with an infection of COVID-19. The results showed that older adults with COVID-19 obtained a higher insomnia severity index when compared with healthy older adults [23].

Overall fear and worry results were evaluated and reported in three studies. Notably, older adults were less worried about the pandemic during the early days of the outbreaks compared to younger groups (*p* < 0.05), and, therefore, paid less attention to precautionary measures than young adults (*p* = 0.004). Nevertheless, as the disease evolved, older adults were more worried than young adults in the second wave and paid more attention to precautionary measures (*p* = 0.027). Living in rural areas, lower BMI index and chronic diseases were considered predictive factors of fear and worry [24]. According to the findings of qualitative interviews in the mixed-method study [35], living in rural areas may suggest very limited information acquisition channels, which can raise older adults’ fear and worry from fake news and invalid information.

### 3.7. Protective Factors against Symptoms of Mental Disorders

Factors that protect Chinese older adults against negative mental health outcomes were also identified. Higher household monthly income per capita, strict community-level entry/exit control, with hobbies, good relationships with children and spouse, and positive aging attitude were all factors that were identified as working against anxiety [24,26,36]. In particular, older adults who are married and not living alone indicated a noteworthy lower depression level (SD = 7.19) when compared with those who live alone (SD = 8.85, *p* = 0.035) [26]. Further, a significant negative correlation between knowledge of the pandemic and anxiety symptoms was identified, indicating that objective and comprehensive understanding of the pandemic may help older adults with good hygiene habits and adequate protection measures, thereby reducing anxiety symptoms [31]. Likewise, actively carrying out precautionary behaviors, such as mask wearing and frequent handwashing also predicted fewer depression symptoms during the pandemic [26]. It must be stressed, however, that though it was not noted as a protective factor, the qualitative interviews also identified that most older adults believe that—under the strong prevention measures of the government and efforts of professional medical staff—the pandemic will be quickly brought under control, and that the country will gradually resume production.

## 4. Discussion

This review investigated the mental health outcomes of Chinese older adults and its associated risk factors during the COVID-19 pandemic. Overall, a greater prevalence of mental health disorders among Chinese older adults was reported when compared to the prevalence before the pandemic [33,34,36]. A wide variation in prevalence between studies were noticed, which could have resulted from distinct measurements tools, different reporting patterns, and time and regional disparity. For instance, while some studies identified only older adults with moderate-to-severe psychological symptoms, others included any older adults with scores above the cut-off point (i.e., mild-to-severe symptoms) [31,36,38]. Regional differences between rural and urban areas existed with respect to Chinese older adults’ mental health conditions during the pandemic due to varying socio-economic levels as well as local government preparedness and measures. Further, the stage of the outbreak also influenced the mental health outcomes of older adults. More severe symptoms of mental disorders were noticed at the early stage of the pandemic when older adults’ lives were faced with unexpected lockdowns [25,28]. Although there is a lack of qualitative studies, the qualitative findings of the only mixed-method study [38] were consistent with the related quantitative findings in other studies. This was achieved by the reporting of the fact that older adults with chronic disease are more likely to show symptoms of hypochondria and living in rural areas may implicate information acquisition channels. Additionally, considering the COVID-19 pandemic is still ongoing in China, further studies are needed in order to continue to assess the mental health outcomes of the COVID-19 pandemic. For example, studies examining the evolution of mental health conditions throughout different periods of the pandemic. Moreover, the strength and resilience of older adults during the pandemic should also be further examined, which may help to identify strategies that are aimed at serving those at higher risk.

### 4.1. Populations with Higher Susceptibility and Psychological Stressors

Female older adults had a higher risk of developing anxiety and depression symptoms [22,34]. On one hand, previous studies indicated that there are sex differences in response to stressors, whereby females initiate stress more rapidly and produces more stress hormones, which may indicate the higher rate of psychological disorders in female older adults [39,40]. On the other hand, female older adults’ daily activities were more strongly affected by quarantine. As they often play the role of caregivers within families, female older adults’ daily activities, such as grocery shopping and caring for grandchildren, may have been greatly impacted. Further, the pandemic may have a greater negative economic effect on females as they represent a greater percentage of the workforce in the manufacturing and retail industry [41]. The exacerbation of the financial situation may bring more stress to female older adults’ mental health, particularly those who are still working in such industries [42].

It was found that the risk of psychological distress in older adults living in rural areas is higher than their urban counterparts [21,24,38]. The reason may be due to the impact of lower socio-economic levels. Older adults living in rural areas encounter relatively more difficulties in basic livelihood security, such as obtaining pandemic prevention materials, medical treatment, and daily necessities. Moreover, as stated in the interview results of the mixed-method study [38], older adults living in rural areas may have very limited ways in which to obtain information about the pandemic. Such situations may expose them to potential false information and, therefore, amplified anxiety.

Contrary to expectations, older adults who participated more in social activities and had good relationships with friends before the pandemic, were at a higher risk of developing anxiety symptoms during the COVID-19 pandemic [21]. The reason for this may be that the quarantines and massive lockdowns greatly affected their social activities and exchanges with friends outside the household [36]. Another hypothesis may be that older adults who participated more in social activities before the pandemic are those who felt lonelier or had less social support at home, which urged them to seek more social support from friends and relatives outside the household [43]. However, the findings also revealed that older adults who are married and live with others were less likely to develop depression and anxiety symptoms [22,26,38]. Such findings may be explained by the fact that older adults living alone may have more difficulties when trying to obtain adequate social support during the pandemic.

Several studies identified that insufficient knowledge relating to COVID-19 and invalid ways of obtaining news about the pandemic, as causes of anxiety and worries [30,31,33]. Invalid ways of dissemination, such as obtaining COVID-19-related information from friends or family members, may expose older adults to possible false information and therefore cause amplified symptoms of anxiety. Further, under the unpredictable situation of the pandemic, fake news and false information are easily spread, which therefore stresses the importance of validated related news/reports.

The studies also indicated that lower education level and lower economic level/household monthly income were associated factors of mental health disorders [22,24,26,30,38]. The COVID-19 pandemic has led to not only large fallbacks in manufacturing and retail industries, but also decreases in demands for all kinds of services and goods [44]. Given such a situation, decreases in family income are common during the pandemic period. However, for those older adults with lower socio-economic levels, the decrease in income may bring financial hardship to the family and put them into greater risk for developing mental health disorders as a result.

### 4.2. Efforts to Cope with Symptoms of Mental Disorders

The risk and protective factors discussed have implications on policy and mental health services aiming to alleviate the negative impact of the pandemic. There is an urgent call for more attention to be paid to the aforementioned vulnerable populations, such as empty nesters and older adults living in rural areas [21,24,38]. Monetary support (e.g., food stamps, unemployment compensation, etc.) should be provided to older adults with financial hardships. Since the validation of information and news related to the pandemic is essential to prevent panic from rumors, the government should also aim to release news/information related to COVID-19, both timely and accurately [45]. Further, an increase in funding allocation to the digital infrastructure in rural areas needs to be provided, along with more official platforms in order to enrich information access channels for older adults in rural areas.

To cope with sudden challenges brought by the pandemic, changes need to be made to ensure the continuity of mental health services. Faced with the challenges of massive lockdowns and quarantines, mental health services adapted by adopting more telehealth methods [46]. Telehealth reduces barriers to access, allows people from remote areas to continue to obtain access to mental healthcare, and is also more cost-effective [47]. Additionally, to better relieve the adverse mental health impacts of the pandemic, more accessible resources, such as psychological behavioral helplines and TV mental health programs, should be provided to older adults in need.

### 4.3. Limitations

This systematic review has limitations. First, due to the high heterogeneity of the assessment tools used as well as the fact that the primary outcomes were measured across studies, a meta-analysis could, therefore, not be conducted. Second, most of the studies were cross-sectional, which limits causal inferences. Third, is that most studies (20 out of 21) utilized self-reported surveys as the method of data collection, which raises the concern that without the supervision of a mental healthcare professional, participants’ response could vary in objectivity. Additionally, 10 studies, which were included, stated that online self-reported surveys were conducted. Thus, older adults without internet accessibility were likely not included in the study, creating a selection bias in the group of general older adults.

## 5. Conclusions

This systematic review investigated the mental health outcomes of the COVID-19 pandemic on Chinese older adults and highlights the associated risk and protective factors related to it. Overall, a higher prevalence (than in pre-pandemic times) of mental health disorders, especially symptoms of anxiety and depression, was reported in most studies. This strongly suggests that the COVID-19 pandemic presents a threat to not only the physical, but also the psychological health of Chinese older adults. As such, government policy and psychological guidelines that are created in order to support older adults’ mental health needs and alleviate the adverse effects of COVID-19, need to be further developed.

## Figures and Tables

**Figure 1 ijerph-19-14362-f001:**
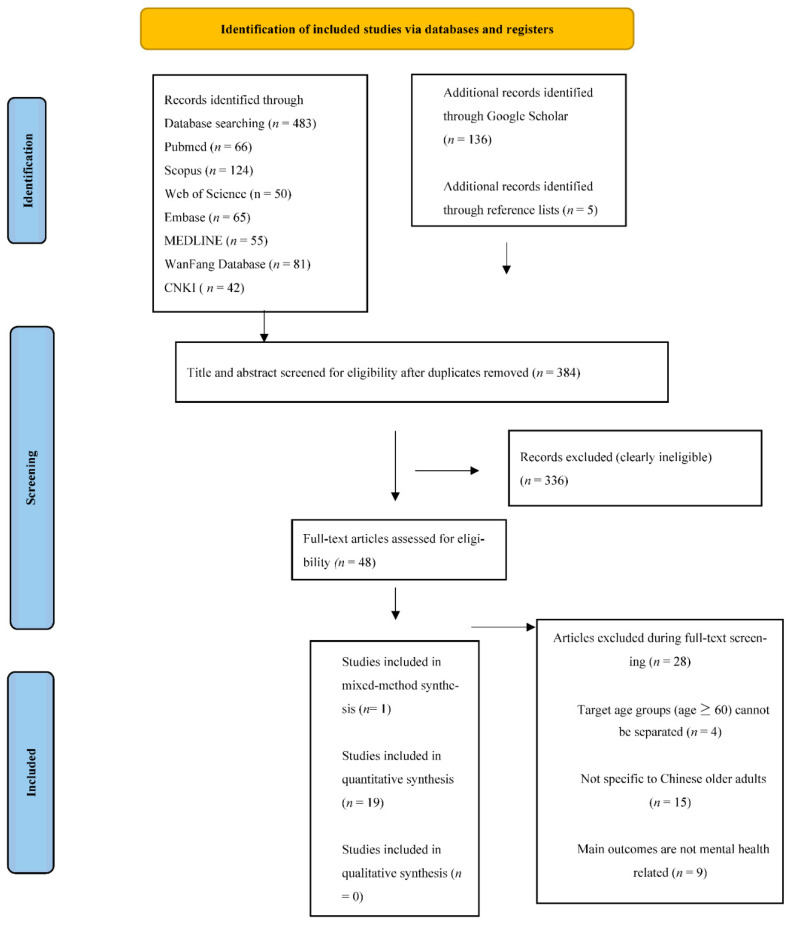
PRISMA flow chart of the study identification process for systematic reviews and meta-analysis, 2019.

**Table 1 ijerph-19-14362-t001:** Major characteristics of studies on the mental health impacts of COVID-19 on Chinese older adults.

Lead Author/Year	Study Design	Sample Size (*n*=)	Sample Characteristics	Sampling Strategy	Data Collection Method	Assessment Tool of Outcomes	Main Outcomes Related to Mental Health
[24]	Cross-sectional study	1278	Female (*n* = 707, 55.3%); Age < 70 (*n* = 725, 56.7%); Primary school or below (*n* = 588, 46%); Rural area dwellers (*n* = 690, 54.0%); Average household income between 600–6000 yuan per month (*n* = 852, 66.7%); Having two or more chronic disease (*n* = 514, 40.2%); Overweight (*n* = 426, 33.3%).	Convenience sampling	Self-reported online survey	PQEFPH	The detective rates of depression, neurasthenia, fear, anxiety, and hypochondria symptoms were 13.8%, 14.9%, 14.7%, 10.0%, and 11.9%, respectively. Chronic disease and the BMI index had a significant effect on all of the five subscales of PQEFPH. The rural dwellers tended to have a greater level of neurasthenia, fear, and hypochondria. Educational level (*p* = 0.035) and outbreak risk level (*p* = 0.004) had significant impacts on instances of depression. Protective factors of anxiety symptoms were higher in participants’ household monthly income per capita (*p* = 0.031) and the community-level entry/exit control.
[20]	Cross-sectional study	49	Age ≥ 60 (*n* = 49, 9.0%) (no other characteristic provided)	Convenience sampling	Self-reported online survey	GSAS	The General Social Alienation Scale Score of 49 participants (aged ≥ 60) is 31.3 ± 5.4, which is higher than the younger group (aged 50–59).
[25]	Cross-sectional study	114	Wave 1: Female (*n* = 874, 76.1%); Age ≥ 60 (*n* = 72, 6.3%); High school or below (*n* = 162, 14.1%), College education or above (*n* = 986, 85.9%). Wave 2: Female (*n* = 357, 76.0%); Age ≥ 60 (*n* = 42, 8.9%); High school or below (*n* = 86, 18.3%), College education or above (*n* = 384, 81.7%).	Convenience sampling	Self-reported online survey	The Worries, Strategies, and Confidence Questionnaire (WSQ)	Compared to younger adults, older adults were less worried about the pandemic at the early stage of the outbreak (*p* < 0.05), and less attention was paid to precautionary measures (*p* = 0.004). However, as the disease evolved, older Chinese adults surveyed in the second wave were more worried than young adults and more attention was paid to precautionary measures (*p* = 0.027).
[26]	Cross-sectional study	516	Female (*n* = 299, 57.9%); Age between 60–69 (*n* = 354, 68.6%); Married (*n* = 432, 83.7%); Primary school or below (*n* = 45, 8.7%); Living with spouse/partners/children (*n* = 468, 90.7%); An average level of household income (*n* = 299, 57.9%); Overweight or obese (*n* = 269, 52.1%).	Snowball sampling	Self-reported online survey	CESD-10	The results stated that the detection rate of depressive symptoms during the pandemic was 30.8%. Factors associated with older adults’ depression levels were living alone, lower education level, lower household income, and infected cases of acquaintances. Precautionary behavior change showed significant inverse associations with older adults’ depression levels, where household income moderated this relationship.
[27]	Cross-sectional study	472	Female (*n* = 266, 56.4%); Age between 60–70 (*n* = 329, 69.7%); Age ≥ 71 (*n* = 142, 30.1%); Married (*n* = 392, 83.1%); High school or above (*n* = 273, 57.8%); Living alone (*n* = 60, 12.7%); Average household income > 5000 yuan per month (*n* = 391, 83.0%).	Quota Sampling	Face to face interview	Life Satisfaction Scale for Chinese Older adults	Cognitive social capital had a mediation impact on the association between structural, social capital, and mental health indicators (life satisfaction: b = 0.122, SD = 0.029, *p* < 0.001; depressive symptoms: b = −0.343, SD = 0.119, *p* < 0.01).
[26]	Cross-sectional study	1159	Female (*n* = 747, 64.5%), Male (*n* = 412, 35.5%); Age between 50–59 (*n* = 867, 74.8%), Age ≥ 60 (*n* = 292, 25.2%); Married (*n* = 1030, 88.9%); Middle school and below (*n* = 256, 22.1%), Associate’s degree and above (*n* = 903, 77.9%); Wuhan residents (*n* = 492, 42.5%), Other places (*n* = 667, 57.5%).	Convenience sampling	Self-reported online survey	A standardized questionnaire adapted from the National Comorbidity Survey	The results showed that 4.1% of participants experienced suicidal ideation during the pandemic. Among those who experienced suicidal ideation, 31.9% believed there is a need for mental health services, yet only 10.6% had reached out for help.
[28]	Cohort study	511	Male (*n* = 176, 34.4%); Age between 60–64 (*n* = 87, 17.0%), Age between 65–69 (*n* = 128, 25.0%), Age between 70–75 (*n* = 141, 27.6%, Age ≥ 76 (*n* = 155, 30.3%).	Convenience sampling	Face to face interview (baseline data T0); Telephone survey (during the outbreak T1; after the subsidence of COVID-19 T2).	PHQ-9	Physical activity and depressive symptoms fluctuated substantially across T0, T1, and T2. Greater depressive symptoms at T0 and T1 were significantly associated with lower levels of physical activity at T1 and T2, but the opposite direction of physical activity being associated with subsequent depressive symptoms was not reported.
[29]	Cross-sectional study	286	Age between 60–74 (*n* = 170, 30.97%), Age ≥ 75 (*n* = 116, 21.13%).	Convenience sampling	Self-reported online survey	GHQ-20	Older adults (aged ≥ 60) showed a significantly greater mean patient health questionnaire anxiety scale (PHQ-20) score (M = 2.65), when compared with participants aged 45–59 years (M = 2.49).
[30]	Cross-sectional study	341	Female (*n* = 195, 57.2%), Male (*n* = 146, 42.8%); Age between 60–70 (*n* = 168, 49.3%), Age between 71–80 (*n* = 136, 39.9%), Age between 81–90 (*n* = 30, 8.8%), Age ≥ 91 (*n* = 7, 2.1%).	Convenience sampling	Self-reported online survey	Self-designed scale, which aimed to investigate older adults’ panic and anxiety about the pandemic, their attitude towards objective measures for epidemic prevention, and their subjective measures for responding to the pandemic.	55.72% of the elderly had obvious fear and worry about COVID-19.
[31]	Cross-sectional study	173	Age ≥ 60 (M = 71.18 ± 6.79)	Random sampling	Self-reported online survey	SAS	A total of 32% of the participants reported different levels of anxiety symptoms, which is much higher than was found in university students (17.5%). There was a significant adverse correlation between participants’ knowledge of the pandemic and anxiety symptoms (*p* < 0.001).
[32]	Cross-sectional study	887	Female (*n* = 582, 65.6%); Age between 60–89 (M = 67.53); High school and below (*n* = 658, 74.18%), College and above (*n* = 229, 25.82%).	Convenience sampling	Telephone survey	Self-designed scale, consisting of seven questions and aiming to test the anxiety symptoms of older adults.	A total of 89.97% (798/887) of the older adults had no obvious anxiety and their mental health has not been significantly affected.
[33]	Cross-sectional study	867	Female (*n* = 378, 43.6%), Male (*n* = 489, 56.4%); Middle school and below (*n* = 413, 47.6%), High school and above (*n* = 454, 52.4%); Married (*n* = 746, 86.0%), Others (*n* = 121, 14.0%); Living alone (*n* = 65, 7.5%), Living with partner only (*n* = 480, 55.4%) Living with family members (include children) (*n* = 322, 37.1%).	Convenience sampling	Self-reported online /tele survey	GDS-30	The detective rate of depression symptoms was increased from 20.9% (181/867) to 27.2% (235/867) during the pandemic (*p* < 0.05). The GDS-30 scores were higher during the pandemic than before the emergencies (9.88 ± 3.85 vs. 7.67 ± 3.54, *p* < 0.05). Predictive factors of depression symptoms were the number of coexisting chronic diseases ≥ 2 (*p* = 0.036), the lack of visits or communication from families (*p* = 0.015), whether the family members represented a risk toward COVID-19 (*p* < 0.05), and the daily viewing of pandemic news for more than 4 h (*p* = 0.023).
[34]	Cross-sectional study	312	Female (*n* = 192, 61.5%), Male (*n* = 120, 38.5%); Age between 65–71 (*n* = 152, 48.7%), Age ≥ 72 (*n* = 160, 51.3%); Married (*n* = 162, 51.9%), Divorced (*n* = 37, 11.9%), Separated living (*n* = 3, 1.0%), Widowed (*n* = 110, 35.3%).	Random sampling	Self-reported survey	PHQ-9 and GAD-7	The detective rates of depression and anxiety symptoms were 28.5% and 27.2%, respectively, which represented a significant increase when compared with the research results before the pandemic (22.11% and 22.6%). The average anxiety scale score of female older adults was greater than male older adults (*p* = 0.014).
[21]	Cross-sectional study	6467 *	Female (*n* = 3599, 55.7%), Male (*n* = 2868, 44.3%); Age between 65–69 (*n* = 2902, 44.9%), Age between 70–74 (*n* = 1867, 28.9%), Age between 75–79 (*n* = 935, 14.5%), Age ≥ 80 (*n* = 763, 11.8%); Rural area dwellers (*n* = 1350, 20.9%), City dwellers (*n* = 5117, 79.1%).	Convenient Sampling	Self-reported survey	GAD-2	The prevalence of anxiety symptoms in the older adults was 4.95% before the COVID pandemic, and 10.10% during the pandemic (*p* < 0.05). Predictive factors of anxiety symptoms were found when living in rural areas, participating in social activities regularly, having a good relationship with friends and being quarantined, or, instead, when the people around were quarantined for medical observation. Protective factors of anxiety symptoms were hobbies, good relationships with children, good relationships with spouse, positive aging attitude, and good psychological resilience (*p* < 0.05).
[35]	Cross-sectional study	235	Female (*n* = 157, 66.8%), Male (*n* = 78, 33.2%); Rural area dwellers (*n* = 69, 29.4%), City dwellers (*n* = 166, 70.6%); Low income (*n* = 101, 43.0%), Medium income (*n* = 123, 52.3%), High income (*n* = 11, 4.7%).	Convenient Sampling	Self-reported online survey	SDS	The detection rate of depressive symptoms was 27.66%. The depression scores of older adults with lower family income were higher than those in the middle and high-income families (F = 7.905, *p* < 0.001).
[36]	Cross-sectional study	320	Female (*n* = 166, 51.9%), Male (*n* = 154, 48.1%); Age between 60–90 (*n* = 297, 92.8%), Age between 90–100 (*n* = 21, 6.6%), Age ≥ 100 (*n* = 2, 0.6%); Primary school and below (*n* = 212, 66.3%), Middle school (*n* = 69, 21.6%), High school and technical secondary school (*n* = 26, 8.1%), College and above (*n* = 13, 4.06%); Rural area dwellers (*n* = 218, 68.1%), City dwellers (*n* = 102, 31.9%).	Convenient Sampling	Self-reported survey	SAS and CD- RISC	The average score of the Self-Rating Anxiety Scale for the elderly was 44.03 ± 10.89, and the detection rate of anxiety was 40.0%. The overall average score of the Connor–Davidson Resilience Scale was 56.68 ± 18.68 points. The predictive factors of anxiety were chronic disease, lower monthly income, and insufficient knowledge about the pandemic.
[37]	Cross-sectional study	6467*	Female (*n* = 3599, 55.7%), Male (*n* = 2868, 44.3%); Age between 65–69 (*n* = 2902, 44.9%), Age between 70–74 (*n* = 1867, 28.9%), Age between 75–79 (*n* = 935, 14.5%), Age ≥ 80 (*n* = 763, 11.8%); Illiteracy (*n* = 870, 13.5%), Primary school (*n* = 2658, 41.1%), Middle school and above (*n* = 2939, 45.4%); Married (*n* = 4944, 76.4%), Others (*n* = 1523, 23.6%); Rural area dwellers (*n* = 1350, 20.9%), City dwellers (*n* = 5117, 79.1%).	Convenient Sampling	Self-reported survey	GAD-2	A total of 576 (9.4%) older adults among 6147 participants without a history of anxiety symptoms were identified as having anxiety symptoms during the pandemic. Two groups of the participants were at a greater risk of having anxiety symptoms, that is: the participants obtaining COVID-19related information from friends or family members (*p* = 0.004) as compared to those obtaining the information from TV/ newspapers; in addition, the participants reporting difficulties in daily life (*p* < 0.001) in comparison with those without such reporting.
[22]	Cross-sectional study	208	Male (*n* = 73, 35.1%), Female (*n* = 135, 64.9%); Age between 60–69 (*n* = 67, 32.2%), Age between 70–79 (*n* = 82, 39.4%), Age ≥ 80 (*n* = 59, 28.4%); Living alone (*n* = 72, 34.6%), Living with people (*n* = 136, 65.4%) High educational level (*n* = 21, 10.1%), Medium educational level (*n* = 84, 40.4%), Low education level (*n* = 103, 49.5%); Monthly income < 2000 yuan (*n* = 61, 29.3%), Monthly income between 2000–5000 yuan (*n* = 132, 63.5%), Monthly income > 5000 yuan (*n* = 15, 7.2%).	Convenient Sampling	Self-reported survey	SAS and SDS	The detective rate of empty nesters’ anxiety symptoms was 48.6% (101/208). The average score of SAS was (46.19 ± 7.78) points, which was higher than the normal population (33.80 ± 5.90, *p* < 0.01). The detective rate of empty nesters’ depression symptoms was 58.7% (122/208), and the average score of SDS was (48.49 ± 10.72) points, which was higher than the normal population (41.88 ± 10.57). The predictive factors of anxiety and depression symptoms were found in low education level, lower monthly income, living alone, older age, and gender (female).
[23]	Cross-sectional study	94	Total: Female (*n* = 36, 38.3%), Male (*n* = 58, 61.7%); COVID-19 Patient Group: Female (*n* = 21, 44.7%), Male (*n* = 26, 55.3%); Age between 62–84 (*n* = 47, M = 73.43); Healthy Group: Female (*n* = 15, 31.9%), Male (*n* = 32, 68.1%); Age between 62–80 (*n* = 47, M = 71.19).	Convenient Sampling	Self-reported survey	SAS, PHQ-9, and ISI	Compared with older adults without the virus (1.48 ± 0.61), the depression symptoms of older adults who had been infected with COVID-19 (1.84 ± 0.75, *p* < 0.05) were significantly higher. The anxiety symptoms between infected older adults and older adults without the virus have no significant difference (*p* > 0.05). The insomnia symptoms of older adults with the virus are significantly more severe than those without the virus (*p* < 0.05).
[38]	Mixed-method study	289	Age between 60–86 (*n* = 289, media*n* = 69.5); Living in the city (*n* = 177, 61.2%), Living in the village (*n* = 112, 38.8%); Living with children (*n* = 144, 49.8%), Living with partner (*n* = 105, 36.4%), Living alone (*n* = 40, 13.8%); Healthy (*n* = 202, 69.9%), Sick (*n* = 87, 30.1%).	Purposive sample (quantitative) and convenient sampling (qualitative)	Self-reported telephone survey and Face to face interview	Self-designed COVID-19 psychological impact questionnaire whereby the questionnaire includes 25 questions that are outlined in five dimensions, which are: depression, neurasthenia, fear, obsessive-compulsive anxiety, and hypochondriasis. The interview outline was based on the basic principles of Van Manen’s deductive phenomenology, which was based on the following questions: (a) Do you know about the new coronavirus?; (b) In the face of the epidemic, what is your greatest feeling?; and (c) Under the new COVID-19 epidemic, what kind of help do you think you need most right now?	The depression symptoms of older adults living in rural areas were more severe than older adults living in urban areas (*p* = 0.015). Depression symptoms of empty nesters (aged > 60) were more severe than other older adults (*p* = 0.035). There was a significant difference in fear and hypochondriasis of older adults with chronic disease and healthy older adults (*p* = 0.028). Qualitative interview results showed insufficient knowledge, which is to say that information acquisition channels are very limited, especially for some older adults living in rural areas. Further, their main information sources are from TV, broadcast, and information from friends. Negative mental health outcomes: (a) Worries: older adults expressed worries about family members being infected and affecting family income. Additionally, some worried about the spread of the pandemic and how it would cause economic loss to the country. (b) Hypochondriasis: during the interview, older adults with chronic diseases were very worried that they would be infected and suspected that they were already infected. Victory belief: (a) have confidence in the country. The country’s strong pandemic prevention system will bring them to win the fight against the virus. (b) have confidence in medical staff.

## Data Availability

All data generated analyzed during this study are included in this published article (and its Appendix A).

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
