# Peer review of "The Mental Health Impact of the COVID-19 Pandemic on Older Adults in China: A Systematic Review"

_ijerph, 2022, doi:10.3390/ijerph192114362_

Round 1

Reviewer 1 Report

The article studied the impact of the new crown pandemic on the mental health of the elderly through the literature method. This study has a strong timeliness and has a certain guiding role for the government to formulate relevant policies. The research content is detailed, the literature research is comprehensive, and the research conclusions are relatively objective and rich. I think this article can be considered for publication.

Author Response

Dear reviewer,

Thank you very much for taking the time to review this manuscript. And we would like to express our great appreciation for your positive comments on our manuscript. 

Best regards,

Jingyuan Liu

Reviewer 2 Report

Pandemic has taken its toll on our mental health globally, affecting different age groups. This review is crucial to identify the issues and to plan for preventive strategies. 

This article is well written, and there are only very minor mistakes detected, which I have highlighted them in the attached file. Suggest to have an english spelling check to ensure there is no careless mistake. 

Thank you. 

Author Response

Thank you very much for taking your time to review this manuscript.  We really appreciate your comments. And we have done a thorough English spelling checking according to your advice. 

Reviewer 3 Report

Thank you for inviting me to review this manuscript. It is an interesting systematic review exploring impact of COVID-19 on older adults’ mental health in China. The authors should be congratulated on conducting this review, however to improve the manuscript I would suggest the following:

ABSTRACT

I suggest removing subtitle for each section and try to keep a whole paragraph for the abstract. Please make sure to be consistent with past tense.

“Greater worries and fear” than? Please make sure to include the comparison.

What about risk factors in the results section?

INTRODUCTION

“To prevent or stop the spread of COVID-19 infection, measures, including stopping production in non-essential business sectors, quarantines and lockdowns, have been taken by authorities in almost all countries around the world”. Please make sure to provide supporting evidence.

It would be helpful to include rates and data specifically on China for COVID pandemic.

I would suggest adding information on pre-pandemic levels of anxiety and mental health issues in China among older adults. This will give a clearer context on how mental health could be impacted before and after the pandemic.

METHODS

Please provide a clearer search strategy in this section. “The search terms were selected based on common synonyms used for each of the main terms of COVID, Chinese, older adults and mental health.” This part and the following are quite vague, please make sure to provide more details to allow the reader to better understand how the search was performed.

Please make sure to justify why the cut-off of 60 years old was used to define older adults and why not the 65 years old one.

Please make sure to be consistent with past tense. i.e. in the strategy for data synthesis subsection, authors are mainly using present. Please make sure to stick with past tense.

Was a search for grey literature performed? If not, why?

Please make sure to justify why the meta-analysis was not performed.

Please provide information on researchers involved in the screening and quality assessment. Were trained researchers? Which background they have?

Please provide information on time. When was the search conducted? The timeframe for the studies to be included which was (i.e. from the start of the pandemic to?)? Please add more details on this.

RESULTS

Subsection 3.4, please make sure to be consistent with the number of decimals (i.e 4.95 and 48.6) I would suggest to the authors sticking with 1 decimal place.

Subsection 3.4, reads as something is missing here “were anxiety” and?

Authors should add information on the grouping for the Synthesis without Meta-analysis.

DISCUSSION

“The reason might be that the quarantines and massive lockdowns have greatly affected their social activities and exchanges with friends outside the household.” Please make sure to provide supporting evidence to this.

I think the conclusion should be better supported by the analysis and results presented. Please make sure to adjust the sections above to support your message.

Author Response

Response to Reviewer’s Comments

We appreciate the reviewer’s detailed suggested and please see our point-to-point replies.

Point 1: ABSTRACT

I suggest removing subtitle for each section and try to keep a whole paragraph for the abstract. Please make sure to be consistent with past tense.

“Greater worries and fear” than? Please make sure to include the comparison.

What about risk factors in the results section?

Response 1

We have removed the subtitle and rechecked it to make sure to be consistent with past tense.

We have modified “greater worries and fear” to “worries and fear” and the detective rate was added.

We have added risk factors in the results section to the abstract. “Female gender, living in rural areas, coexisting chronic diseases and insufficient knowledge about the COVID pandemic were most common risk factors of phycological distress among Chinese older adults.”

Point 2: INTRODUCTION

“To prevent or stop the spread of COVID-19 infection, measures, including stopping production in non-essential business sectors, quarantines and lockdowns, have been taken by authorities in almost all countries around the world”. Please make sure to provide supporting evidence.

It would be helpful to include rates and data specifically on China for COVID pandemic.

I would suggest adding information on pre-pandemic levels of anxiety and mental health issues in China among older adults. This will give a clearer context on how mental health could be impacted before and after the pandemic.

Response 2:

We have added supporting evidence and the citation (Douglas, 2022) accordingly.

We have added COVID-related death rates of older adults in Shanghai accordingly. “In China, during the pandemic,more than 90% of COVID-related deaths in Shanghai were older adults ( Shanghai Municipal Health Commission, 2022).”

We have demonstrated the comparison of pre-pandemic levels and after the pandemic of anxiety and other mental health issues in the Chinese older adults in table 1 (e.g., “The prevalence of anxiety symptoms in the older adults was 4.95% before the COVID pandemic, and 10.10% during the pandemic (P<0.05)”) and the results section (e.g., “According to Ye and Lin (2021), participants’ (n = 340) detection rate of anxiety symptoms had significantly increased from 22.6% to 27.2% (p < 0.05), compared with previous research results before the pandemic”), which help to illustrate and better explain the impact of the pandemic.

Point3: METHODS

Please provide a clearer search strategy in this section. “The search terms were selected based on common synonyms used for each of the main terms of COVID, Chinese, older adults and mental health.” This part and the following are quite vague, please make sure to provide more details to allow the reader to better understand how the search was performed.

Please make sure to justify why the cut-off of 60 years old was used to define older adults and why not the 65 years old one.

Please make sure to be consistent with past tense. i.e. in the strategy for data synthesis subsection, authors are mainly using present. Please make sure to stick with past tense.

Response 3:

The following is detailed in the methods section regarding common synonyms:

“The search terms were selected based on common synonyms used for each of the main terms of COVID, Chinese, older adults and mental health. Related words, synonyms and plurals of the main terms were selected according to the search in databases like PubMed. The team also identified synonyms or related words based on other published systematic reviews that focused on mental health impact of COVID-19 (e.g., Nina & Mi-chael, 2020). The COVID terms included COVID* OR COVID-19 OR Coronavirus. The mental health terms included mental health* OR mental disorders OR psychological stress OR psychiatric symptoms OR mental illness. The older adults’ terms included elderly* OR senior OR older adults* OR older people OR older persons OR aging population OR aged people.”

We used 60 years old as the cut-off point as it aligns with the United Nations definition (UNHCR,2017). We have modified and rechecked the tense accordingly.

Point 4: METHODS

Was a search for grey literature performed? If not, why?

Please make sure to justify why the meta-analysis was not performed.

Please provide information on researchers involved in the screening and quality assessment. Were trained researchers? Which background they have?

Please provide information on time. When was the search conducted? The timeframe for the studies to be included which was (i.e. from the start of the pandemic to?)? Please add more details on this

Response 4:

A search for grey literature was not performed as the aim of the review was to focus on rigorous empirical studies, and peer-reviewed research is one initial way to filter the rigor of the studies. Also, the channels and resources of accessing unpublished studies were very limited, which confined us from doing a thorough search for grey literature.

We justified the reason of why a meta-analysis could not be conducted in the methods section: “Since the studies included were heterogenous, Synthesis without meta-analysis (SWiM) by Campbell et al. (2019) was planned for the quantitative findings.”

 All the researchers involved in the screening and quality assessment are trained researchers. DJ and HYX are research assistants in the Hong Kong Polytechnic University, with Master’s degree of Phycology. JL and CK is a doctoral student and assistant professor in social work, respectively.   

We have added the specific time when the search was conducted from Month Year to Month Year; and specific timeframe to be included for the studies January 2020 to April 2022.

Point 5: RESULTS

Subsection 3.4, please make sure to be consistent with the number of decimals (i.e 4.95 and 48.6) I would suggest to the authors sticking with 1 decimal place.

Subsection 3.4, reads as something is missing here “were anxiety” and?

Authors should add information on the grouping for the Synthesis without Meta-analysis.

Response 5:

We have Revised all decimals to be consistently within 1 decimal place.

We have revised the sentence to the following:

“The most commonly evaluated mental health outcome was anxiety (n = 10), with the prevalence varying from 4.9 % to 48.6% between studies (Lu et al., 2021; Liu & Liu., 2021; Qi et al., 2021; Su et al., 2021; Tao et al., 2021; Zhou et al., 2021; Su et al., 2021; Ye & Lin., 2021; Wang et al., 2020; Wang et al., 2021; Xue et al., 2021).”

The following is the information on the grouping for the Synthesis:

There were no qualitative studies and only one mixed-method study. The quantitative findings of the mixed-method study were extracted and synthesized with findings of other quantitative studies. The integration of qualitative and quantitative findings was guided by Creswell and Plano-Clark (2018).”

Point 6: DISCUSSION

“The reason might be that the quarantines and massive lockdowns have greatly affected their social activities and exchanges with friends outside the household.” Please make sure to provide supporting evidence to this.

I think the conclusion should be better supported by the analysis and results presented. Please make sure to adjust the sections above to support your message.

Response 6:

We have added a citation (Wang et al., 2021) to support the study accordingly.

Within the discussion section we explicitly linked the points to the results and analysis, with explicit references to our study findings and analysis. The following are examples of direct links/references with the included studies within the discussion section. For example, in paragraph 1:

“Overall, A greater prevalence of mental health disorders among Chinese older adults was reported when compared to the prevalence before the pandemic (Bao et al., 2021; Wang et al., 2021; Ye & Lin, 2021;).”

Other examples are in 4.1:

“Female older adults had a higher risk of developing anxiety and depressive symptoms (Ye & Lin, 2021; Xue et al., 2021).”

“It was found that the risk of psychological distress in older adults living in rural areas is higher than their urban counterparts (Xu et al., 2021; Zhou et al, 2021; Wang et al., 2020).”

“Several studies identified insufficient knowledge relating to COVID-19 and invalid ways of obtaining news about the pandemic as causes of anxiety and worries (Bao et al., 2021; Tao et al., 2021; Wang et al; 2021;)."

Another example is in 4.2:

“There is an urgent call for more attention paid to the aforementioned vulnerable populations, such as empty nesters and older adults living in rural areas (Xu et al., 2021; Zhou et al, 2021; Wang et al., 2020)."

Round 2

Reviewer 3 Report

Thanks for all your answers. I'm happy with the current version of the paper.